# Gyrification of the cerebral cortex requires FGF signaling in the mammalian brain

**Naoyuki Matsumoto, Yohei Shinmyo, Yoshie Ichikawa, Hiroshi Kawasaki***

Department of Medical Neuroscience, Graduate School of Medical Sciences, Kanazawa University, Ishikawa, Japan

**Abstract** Although it has been believed that the evolution of cortical folds was a milestone, allowing for an increase in the number of neurons in the cerebral cortex, the mechanisms underlying the formation of cortical folds are largely unknown. Here we show regional differences in the expression of fibroblast growth factor receptors (FGFRs) in the developing cerebral cortex of ferrets even before cortical folds are formed. By taking the advantage of our *in utero* electroporation technique for ferrets, we found that cortical folding was impaired in the ferret cerebral cortex when FGF signaling was inhibited. We also found that FGF signaling was crucial for producing Pax6-positive neural progenitors in the outer subventricular zone (OSVZ) of the developing cerebral cortex. Furthermore, we found that upper layers of the cerebral cortex were preferentially reduced by inhibiting FGF signaling. Our results shed light on the mechanisms of cortical folding in gyrencephalic mammalian brains.
DOI: https://doi.org/10.7554/eLife.29285.001

## Introduction

Folds of the cerebral cortex, which are called the gyri and the sulci, are one of the most prominent features of the mammalian brain. Humans, monkeys and ferrets have gyrencephalic brains (i.e. brains with folded cerebral cortices), whereas the brains of rodents are often lissencephalic (i.e. lacking cortical folds). It has been believed that the creation of cortical folds during evolution was a milestone, allowing for an increase in the number of neurons in the cerebral cortex (*Lewitus et al., 2013*; *Sun and Hevner, 2014*). Malformations of cortical folds in human patients are associated with severe intellectual disabilities, epilepsy and diseases such as lissencephaly, polymicrogyria, schizophrenia and autism (*Ross and Walsh, 2001*). Therefore, the molecular and cellular mechanisms underlying the formation and malformation of cortical folds during development have been of great interest; however, they are still largely unknown.

One attractive hypothesis for the mechanism of cortical folding is that an increase in the numbers of neural progenitors is responsible for cortical folding (*Kriegstein et al., 2006*; *Molnár et al., 2006*; *Rakic, 1995*; *Fietz and Huttner, 2011*; *Rakic, 2009*; *Lui et al., 2011*; *Poluch and Juliano, 2015*; *Sun and Hevner, 2014*; *Borrell and Götz, 2014*; *Dehay et al., 2015*). The developing cerebral cortex contains two germinal layers containing neural progenitors: the ventricular zone (VZ) and the subventricular zone (SVZ). The VZ comprises radial glial cells (RG cells, also known as apical progenitors/ventricular RG cells/apical RG cells), which undergo multiple rounds of cell divisions and generate SVZ progenitors. The SVZ is further subdivided into the outer SVZ (OSVZ) and the inner SVZ (ISVZ), and contains two types of neural progenitors: intermediate progenitor cells (IP cells, also known as basal progenitors) and outer radial glial cells (oRG cells, also known as OSVZ RG cells/basal RG cells/intermediate RG cells/translocating RG cells). Because a prominent thick SVZ was found in gyrencephalic animals, it has been hypothesized that thickening of the SVZ leads to

*For correspondence:
hiroshi-kawasaki@umin.ac.jp

Competing interests: The authors declare that no competing interests exist.

acquiring cortical folds during development and evolution (*Martínez-Cerdeño et al., 2006*; *Fietz et al., 2010*; *Hansen et al., 2010*; *Reillo and Borrell, 2012*). However, the hypotheses about the mechanisms of cortical folding had been difficult to test experimentally in vivo, mainly because rapid and efficient genetic manipulation techniques that can be applied to gyrencephalic mammalian brains were poorly available.

To overcome this difficulty, we recently developed a genetic manipulation technique for gyrencephalic carnivore ferrets using *in utero* electroporation (IUE) (*Kawasaki et al., 2012*, *2013*). By taking the advantage of our IUE technique for ferrets, we successfully demonstrated that the transcription factor Tbr2 was required for producing SVZ progenitors and that SVZ progenitors were indeed crucial for cortical folding during development (*Toda et al., 2016*). These results uncovered cell-autonomous molecular mechanisms of cortical folding. The next important question was what the upstream mechanisms regulating SVZ production and cortical folding were. Recently, we succeeded in producing a ferret model of polymicrogyria (*Masuda et al., 2015*). When fibroblast growth factor (FGF) was overexpressed into the ferret cerebral cortex using our IUE technique, a number of undulating folds appeared, suggesting that excess FGF is sufficient for producing additional cortical folds (*Masuda et al., 2015*). This finding further raised the possibility that endogenous FGF signaling is crucial for cortical folding during development. Here we uncovered that cortical folding was impaired in the ferret cortex when FGF signaling was inhibited. We further found that Pax6-positive oRG cells and upper layers of the cerebral cortex were selectively reduced by inhibiting FGF signaling. Our results shed light on the mechanisms of cortical folding in gyrencephalic mammalian brains.

## Results

### FGF signaling mediates the formation of cortical folds

We first examined which FGF receptors (FGFRs) are expressed in the developing cerebral cortex of ferrets at postnatal day 0 (P0), when cortical folds are about to be formed. Reverse transcription-PCR (RT-PCR) demonstrated that *FGFR1*, *FGFR2* and *FGFR3* were expressed in the ferret cerebral cortex (*Figure 1A*). To examine the role of FGF signaling in cortical folding, we inhibited FGF signaling in the developing ferret cortex. Because multiple FGFRs are expressed in the cortex, we introduced a soluble extracellular domain of FGFR3 (sFGFR3), which acts in a dominant-negative manner, into the ferret cerebral cortex using our IUE technique to suppress all FGFR activity (*Figure 1B*). It should be noted that sFGFR3, which consists of the extracellular domain of FGFR3 without the transmembrane domain, is released from transfected cells and binds to endogenous FGFs (*Fukuchi-Shimogori and Grove, 2001*). It was therefore supposed that sFGFR3 suppresses FGF signaling not only in transfected cells but also in neighboring non-transfected cells non-cell-autonomously, presumably resulting in stronger phenotypes. To test this, we introduced sFGFR3 into the ferret cerebral cortex using IUE and performed in situ hybridization for *Sprouty2*, whose expression is known to be up-regulated by FGFR activation (*Tsang and Dawid, 2004*). *Sprouty2* signals were abundantly distributed in the OSVZ and the VZ of the control ferret cortex (*Figure 1—figure supplement 1*). In contrast, *Sprouty2* signals in non-transfected cells, which did not show GFP signals in their somata, were markedly suppressed by sFGFR3 electroporation in the OSVZ (*Figure 1—figure supplement 1*). These results suggest that sFGFR3 indeed suppresses FGF signaling non-cell-autonomously.

We introduced sFGFR3 into the ferret cortex using IUE at embryonic day 33 (E33) and examined cortical folds at P16. Interestingly, we found that cortical folding was impaired in sFGFR3-transfected areas (*Figure 1B*). Consistently, coronal sections showed that cortical folding was inhibited in the sFGFR3-transfected side of the cortex compared with the contralateral side of the cortex and the EGFP-transfected control cortex (*Figure 1C,D*). To quantify the effects of sFGFR3 on cortical folding, we utilized the local gyrus size (GS) ratio (*Figure 1—figure supplement 2*), the local sulcus depth (SD) ratio (*Figure 1—figure supplement 3*) and the local gyrification index (GI) ratio (*Figure 1—figure supplement 4*). Consistent with our observation, the local GS ratio, the local SD ratio and the local GI ratio were significantly reduced by sFGFR3 (local GS ratio: control, $0.75 \pm 0.05$; sFGFR3, $0.25 \pm 0.05$; p=0.003; Student's *t*-test) (local SD ratio: control, $0.88 \pm 0.03$; sFGFR3, $0.47 \pm 0.10$; p=0.019; Student's *t*-test) (local GI ratio: control, $0.95 \pm 0.04$; sFGFR3, $0.72 \pm 0.05$; p=0.025; Student's *t*-test) (*Figure 1E–G*, *Supplementary file 1*). These results clearly indicate that FGF signaling is crucial for cortical folding. It should be noted that cortical folding in peripheral regions of the

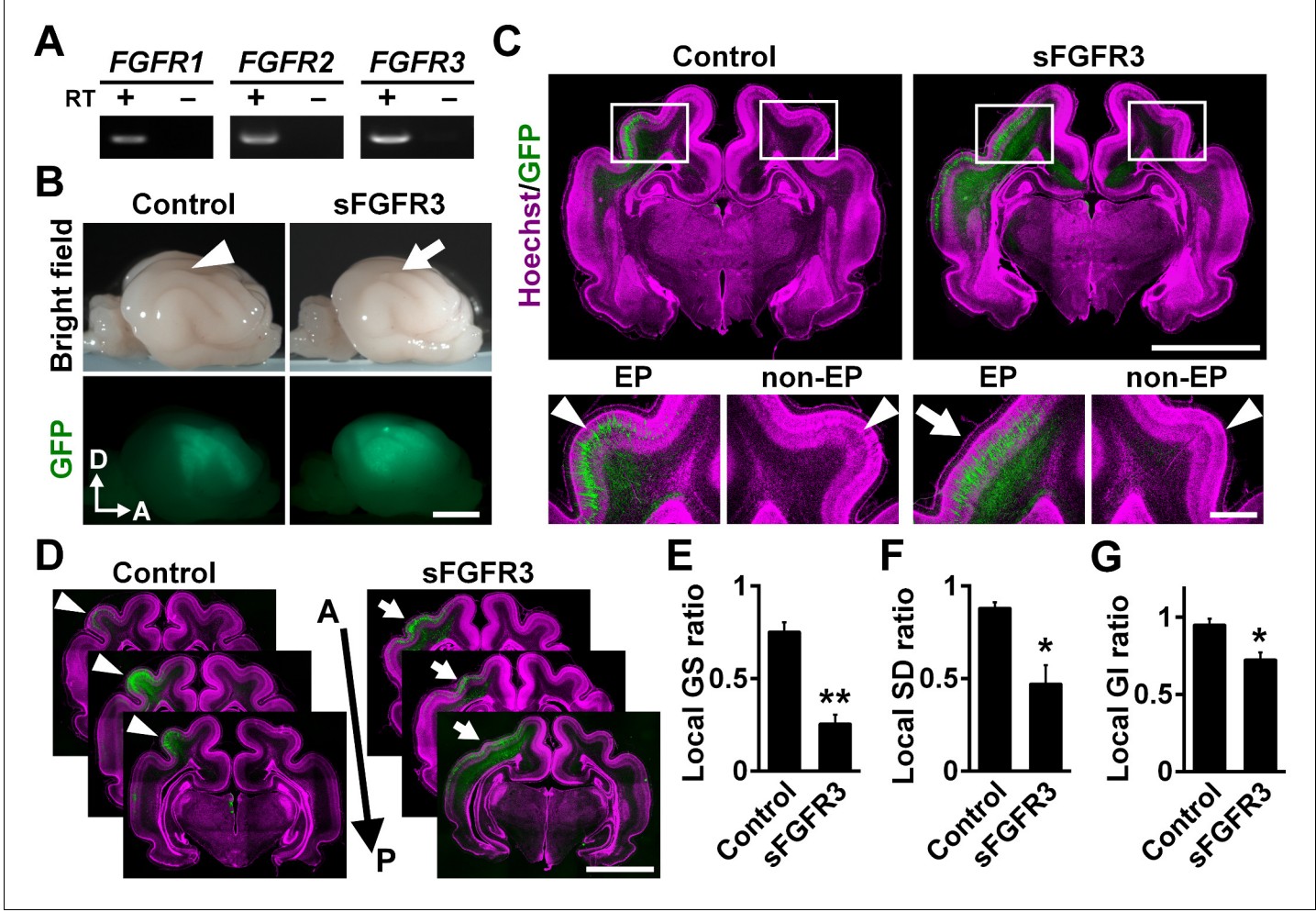

**Figure 1.** FGF signaling is required for cortical folding. (**A**) Expression of FGFRs in the developing ferret cortex. RT-PCR was performed using the cerebral cortex of the ferret at P0. RT+ and RT- indicate samples with and without reverse transcriptase, respectively. (**B**) Lateral views of the transfected brains. pCAG-EGFP plus either pCAG-sFGFR3 or pCAG control vector was electroporated at E33, and the brains were dissected at P16. Note that sFGFR3 suppressed the formation of the gyrus (arrow), which was present in the control brain (arrowhead). A, anterior; D, dorsal. Scale bar = 5 mm. (**C**) Coronal sections of the brain stained with anti-GFP antibody and Hoechst 33342 (magenta). The formation of cortical folds was suppressed in the GFP-positive area when sFGFR3 was electroporated (sFGFR3, EP, arrow). Arrowheads indicate cortical folds in the control cortex and in the non-electroporated side of the cortex (non-EP). Scale bars = 5 mm (upper panel) and 1 mm (lower panel). (**D**) Serial coronal sections of the brain stained with anti-GFP antibody and Hoechst 33342 (magenta). The formation of cortical folds was widely suppressed in the GFP-positive areas when sFGFR3 was transfected (arrows). Arrowheads indicate cortical folds in the control cortex. A, anterior; P, posterior. Scale bar = 5 mm. (**E–G**) Quantification of the local GS ratio (**E**), the local SD ratio (**F**) and the local GI ratio (**G**). The local GS ratio, the local SD ratio and the local GI ratio were significantly smaller in sFGFR3-transfected samples. n = 3 animals for each condition. Bars present mean ± SEM. *p<0.05, **p<0.01, Student's *t*-test.

DOI: https://doi.org/10.7554/eLife.29285.002

The following figure supplements are available for figure 1:

**Figure supplement 1.** sFGFR3 suppressed FGF signaling in the OSVZ in a non-cell-autonomous manner.

DOI: https://doi.org/10.7554/eLife.29285.003

**Figure supplement 2.** The definition of the local gyrus size (GS) ratio.

DOI: https://doi.org/10.7554/eLife.29285.004

**Figure supplement 3.** The definition of the local sulcus depth (SD) ratio.

DOI: https://doi.org/10.7554/eLife.29285.005

**Figure supplement 4.** The definition of the local gyrification index (GI) ratio.

DOI: https://doi.org/10.7554/eLife.29285.006

GFP-positive area did not seem to be affected in the sFGFR3-transfected cortex. It seemed likely that the expression level of sFGFR3 was not sufficient to inhibit FGF signaling strongly in peripheral regions of the GFP-positive area. We also noticed that the local GS ratio and the local SD ratio were slightly smaller than 1 in the GFP-transfected control cortex (local GS ratio, p=0.009; local SD ratio, p=0.02; local GI ratio, p=0.19; one sample $t$-test), raising the possibility that cortical folding can be affected by electroporation itself. It would be important to be careful when interpreting experimental results using IUE. In addition, it is worth noting that because GFP-positive areas were wide enough to include both the gyrus and sulcus regions, it seemed possible that sFGFR3 affected both the formation of the gyrus and that of the sulcus.

## Regional differences in FGFR expression in the ferret cerebral cortex during development

We examined the detailed expression patterns of *FGFR1*, *FGFR2* and *FGFR3* in coronal sections of the ferret cerebral cortex during development. In situ hybridization showed that *FGFR2* and *FGFR3* mRNAs were predominantly expressed in the germinal zones such as the VZ at E32, and the VZ and the SVZ at E36 and E40 (*Figure 2—figure supplement 1A–C*). *FGFR1* was also expressed in the VZ, but its signals were weaker than those of *FGFR2* and *FGFR3* (*Figure 2—figure supplement 1A–C*). Interestingly, the expression of *FGFR2* and that of *FGFR3* in the OSVZ were not uniform throughout the cortex at E36 and E40. Some areas contained abundant *FGFR* signals in the OSVZ, whereas other areas had fewer *FGFR* signals (*Figure 2—figure supplement 1B,C*). These results led to the hypothesis that the regional differences in FGFR expression in the OSVZ mediate cortical folding. We quantified *FGFR* signal intensities along the tangential axis in the OSVZ at E40 (*Figure 2—figure supplement 1D–G*). *FGFR* signals indeed showed regional differences in the OSVZ (*Figure 2—figure supplement 1D–G*).

Previously, lateral to medial gradients in cell proliferation were reported in the rodent cerebral cortex (*Bayer and Altman, 1991*). Similarly, it was also reported that *Fgfr* expression showed medio-lateral gradients in the VZ of the rodent cerebral cortex during early corticogenesis, although it became uniform afterward (*Hasegawa et al., 2004*; *Hébert et al., 2003*). It therefore seemed possible that the regional differences in *FGFR* signals found in the ferret cortex reflected the medio-lateral gradients found in mice. Another possibility was that the regional differences in *FGFR* expression in ferrets corresponded to the positions of gyri and sulci. To address these possibilities, we next performed in situ hybridization for *FGFR2* and *FGFR3* at P6, when the gyrus was beginning to be formed. We found that *FGFR2* and *FGFR3* were abundantly expressed in the region where the gyrus would be formed, but less abundant where the sulcus would be formed (*Figure 2—figure supplement 2*). These results suggest that the regional difference in *FGFR* expression in the OSVZ corresponds to the positions of the gyrus and the sulcus. Our findings were consistent with a previous work showing uneven distribution patterns of *FGFR2* and *FGFR3* in the ferret cortex (*de Juan Romero et al., 2015*). In contrast, there were no similar regional differences in *Fgfr* expression in the lissencephalic cerebral cortex of developing mice at E15, which corresponds to E40 in ferrets (*Figure 2—figure supplement 3*) (*Bansal et al., 2003*). The regional differences in *FGFR* expression in the OSVZ during development therefore seemed to be an important feature of the gyrencephalic cortex of mammals.

Previous studies uncovered regional differences in the distribution of SVZ progenitors in the developing ferret cerebral cortex (*Toda et al., 2016*; *de Juan Romero et al., 2015*) and reported that the regional differences of SVZ progenitors were correlated with the positions of gyri (*Toda et al., 2016*; *Reillo et al., 2011*; *Nowakowski et al., 2016*; *Smart et al., 2002*). It seemed possible that the regional differences of *FGFRs* correspond to those of SVZ progenitors. We therefore examined the expression pattern of Pax6, which is expressed in RG cells and oRG cells, and found that *FGFRs* and Pax6 showed similar regional differences in the OSVZ (*Figure 2—figure supplement 1B,C*). These findings are consistent with the idea that FGF signaling in the OSVZ underlies cortical folding of the ferret cerebral cortex.

## Identities of *FGFR*-positive cells in the OSVZ

To determine the cellular identities of *FGFR*-positive cells in the OSVZ at a single-cell level, we performed triple staining with in situ hybridization for *FGFRs* and immunostaining for Pax6 and Tbr2.

We found that *FGFR2*-positive cells and *FGFR3*-positive cells in the OSVZ were mostly positive for Pax6 (*Figure 2A*, arrows) but negative for Tbr2 (*Figure 2B*, arrowheads and 2C) (*FGFR2*: Pax6+/Tbr2- cells, 91.4 ± 6.2%; Pax6+/Tbr2+ cells, 3.8 ± 3.1%; Pax6-/Tbr2+ cells, 0.0%; Pax6-/Tbr2- cells, 4.9 ± 4.4%) (*FGFR3*: Pax6+/Tbr2- cells, 76.7 ± 5.7%; Pax6+/Tbr2+ cells, 12.4 ± 5.7%; Pax6-/Tbr2+ cells, 0.0%; Pax6-/Tbr2- cells, 10.8 ± 10.5%). Similar results were obtained for *FGFR1* (Pax6+/Tbr2- cells, 78.7 ± 10.0%; Pax6+/Tbr2+ cells, 5.8 ± 5.4%; Pax6-/Tbr2+ cells, 0.0%; Pax6-/Tbr2- cells, 15.4 ± 6.5%) (*Figure 2A*, arrows; 2B, arrowheads and 2C), although *FGFR1*-positive cells were fewer. Because it was previously reported that astrocytes also expressed Pax6 in the mouse cerebral cortex (*Sakurai and Osumi, 2008*), we next examined whether Pax6-positive cells in the OSVZ included astrocytes. We found that Pax6-positive cells in the OSVZ did not express GFAP (*Figure 2—figure supplement 4*). These results suggest that *FGFR*-positive cells in the OSVZ are mostly Pax6-positive/Tbr2-negative oRG cells rather than Tbr2-positive IP cells or astrocytes.

## FGF signaling is indispensable for oRG production

Because FGFRs are expressed in oRG cells, we next examined the role of FGF signaling in the production of oRG cells by expressing sFGFR3. We found that sFGFR3 markedly reduced Pax6-positive cells in the OSVZ underneath EGFP-positive transfected areas at P6 (*Figure 3B–D*). Our quantification showed that Pax6-positive cells in the OSVZ were significantly decreased by sFGFR3 (OSVZ: Control, 9.40 ± 2.25; sFGFR3, 2.54 ± 0.77; p=0.008; Student's *t*-test), while those in the VZ and the ISVZ were not (VZ: Control, 8.70 ± 2.30; sFGFR3, 11.12 ± 1.71; p=0.15; Student's *t*-test) (ISVZ: Control, 17.9 ± 7.1; sFGFR3, 13.4 ± 4.7; p=0.25; Student's *t*-test) (*Figure 3E*, *Supplementary file 1*). Interestingly, Tbr2-positive cells were also reduced by sFGFR3 in the OSVZ selectively (VZ: Control, 1.96 ± 0.80; sFGFR3, 2.02 ± 0.29; p=0.46; Student's *t*-test) (ISVZ: Control, 8.79 ± 3.54; sFGFR3, 6.74 ± 2.41; p=0.27; Student's *t*-test) (OSVZ: Control, 6.37 ± 2.08; sFGFR3, 2.26 ± 0.32; p=0.025; Student's *t*-test) (*Figure 3F,G*, *Supplementary file 1*), although FGFR was rarely expressed in Tbr2-positive cells (*Figure 2*). We further examined the detailed identities of the affected cells in the OSVZ using double immunostaining for Pax6 and Tbr2. We found that both Pax6-positve/Tbr2-negative oRG cells and Pax6-positive/Tbr2-positive IP cells were significantly decreased by sFGFR3 in the OSVZ (Pax6-positive/Tbr2-negative cells: Control, 2.38 ± 0.41 per 1000 $\mu m^2$; sFGFR3, 1.61 ± 0.26 per 1000 $\mu m^2$; p=0.044; Student's *t*-test) (Pax6-positive/Tbr2-positive cells: Control, 1.49 ± 0.69 per 1000 $\mu m^2$; sFGFR3, 0.35 ± 0.13 per 1000 $\mu m^2$; p=0.041; Student's *t*-test) (*Figure 3—figure supplement 1*, *Supplementary file 1*). These results suggest that FGF signaling is essential for producing OSVZ progenitors. Because a previous study showed that oRG cells in OSVZ have the ability to produce IP cells (*Hansen et al., 2010*), it seemed plausible that reduction of Pax6-positive/Tbr2-negative oRG cells resulted in that of Tbr2-positive cells.

We next tested whether cell proliferation in the OSVZ was affected by sFGFR3. As expected, we found that Ki-67-positive cells in the OSVZ, but not in the VZ and the ISVZ, were significantly reduced by sFGFR3 (VZ: Control, 3.22 ± 0.38 per 1000 $\mu m^2$; sFGFR3, 3.07 ± 0.38 per 1000 $\mu m^2$; p=0.36; Student's *t*-test) (ISVZ: Control, 2.15 ± 0.55 per 1000 $\mu m^2$; sFGFR3, 2.19 ± 1.07 per 1000 $\mu m^2$; p=0.48; Student's *t*-test) (OSVZ: Control, 1.75 ± 0.08 per 1000 $\mu m^2$; sFGFR3, 0.76 ± 0.28 per 1000 $\mu m^2$; p=0.004; Student's *t*-test) (*Figure 3—figure supplement 2A,B*, *Supplementary file 1*). Consistent results were obtained using anti-phospho-histone H3 (pHH3) (VZ: Control, 0.28 ± 0.02 per 1000 $\mu m^2$; sFGFR3, 0.29 ± 0.07 per 1000 $\mu m^2$; p=0.39; Student's *t*-test) (ISVZ: Control, 0.28 ± 0.08 per 1000 $\mu m^2$; sFGFR3, 0.23 ± 0.06 per 1000 $\mu m^2$; p=0.26; Student's *t*-test) (OSVZ: Control, 0.091 ± 0.041 per 1000 $\mu m^2$; sFGFR3, 0.024 ± 0.008 per 1000 $\mu m^2$; p=0.043; Student's *t*-test) and anti-phosphorylated vimentin (pVim) (VZ: Control, 0.57 ± 0.33 per 1000 $\mu m^2$; sFGFR3, 0.69 ± 0.07 per 1000 $\mu m^2$; p=0.33; Student's *t*-test) (ISVZ: Control, 0.20 ± 0.10 per 1000 $\mu m^2$; sFGFR3, 0.34 ± 0.15 per 1000 $\mu m^2$; p=0.16; Student's *t*-test) (OSVZ: Control, 0.096 ± 0.032 per 1000 $\mu m^2$; sFGFR3, 0.038 ± 0.020 per 1000 $\mu m^2$; p=0.049; Student's *t*-test) antibodies (*Figure 3—figure supplement 2C–F*, *Supplementary file 1*). These results indicate that the number of proliferating cells in the OSVZ was reduced by sFGFR3.

We then examined which cell types had their proliferation affected by sFGFR3 in the OSVZ. We performed triple immunostaining for Pax6, Tbr2 and pHH3, and found that the percentage of Pax6-positive/Tbr2-negative cells co-expressing pHH3 in the OSVZ was significantly reduced by sFGFR3 (Control, 2.57 ± 0.91%; sFGFR3, 0.67% ± 0.34%; p=0.025; Student's *t*-test), while that of Pax6/Tbr2-double positive cells co-expressing pHH3 was not (Control, 1.28 ± 0.60%; sFGFR3, 0.89% ± 1.25%;

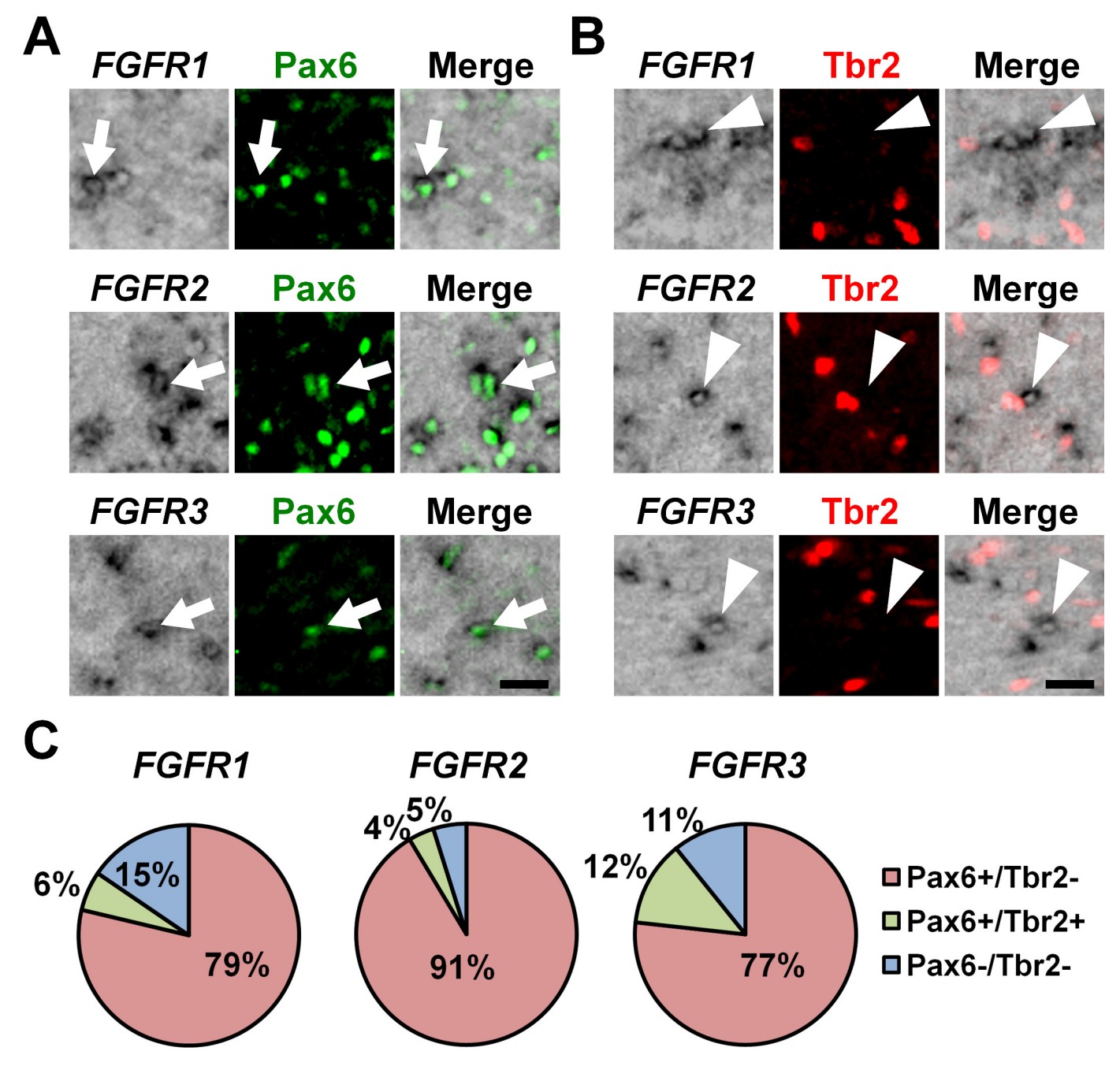

**Figure 2.** The expression of Pax6 and Tbr2 in *FGFR*-positive cells in the OSVZ. Sections of the developing ferret cortex at P6 were subjected to immunohistochemistry for Pax6 or Tbr2 and in situ hybridization for *FGFR1*, *FGFR2* or *FGFR3*. (**A and B**) Higher magnification images of the OSVZ are shown. Note that *FGFR*-positive cells were mostly positive for Pax6 (arrows), but negative for Tbr2 (arrowheads). Scale bars = 50 μm. (**C**) Quantification of the percentages of *FGFR*-positive cells which were also positive for Pax6 and Tbr2. n = 3 animals for each condition. For details, see *Supplementary file 1*.

DOI: https://doi.org/10.7554/eLife.29285.007

The following figure supplements are available for figure 2:

**Figure supplement 1.** Regional differences in the abundance of *FGFR* expression in the developing cerebral cortex of ferrets.

DOI: https://doi.org/10.7554/eLife.29285.008

**Figure supplement 2.** Regional differences in the abundance of *FGFR* expression in the postnatal ferret cerebral cortex.

DOI: https://doi.org/10.7554/eLife.29285.009

*Figure 2 continued on next page*

*Figure 2 continued*

**Figure supplement 3.** The expression of *Fgfr1*, *Fgfr2* and *Fgfr3* in the developing mouse cerebral cortex.
DOI: https://doi.org/10.7554/eLife.29285.010

**Figure supplement 4.** Pax6-positive cells in the OSVZ did not express GFAP in the developing cerebral cortex of ferret neonates.
DOI: https://doi.org/10.7554/eLife.29285.011

p=0.36; Student's *t*-test) (*Figure 4*, *Supplementary file 1*). Thus, our findings indicate that proliferation of Pax6-positive/Tbr2-negative oRG cells in the OSVZ is regulated by FGF signaling. This is consistent with our data that FGFRs are expressed in Pax6-positive/Tbr2-negative cells, but not in Tbr2-positive cells in the OSVZ (*Figure 2*). In addition, we also examined apoptosis using anti-cleaved caspase-3 antibody, but did not find significant effects of sFGFR3 on the number of cleaved caspase-3-positive cells (VZ: Control, $0.27 \pm 0.07$ per 1000 $\mu m^2$; sFGFR3, $0.42 \pm 0.23$ per 1000 $\mu m^2$; p=0.23; Student's *t*-test) (ISVZ: Control, $0.22 \pm 0.06$ per 1000 $\mu m^2$; sFGFR3, $0.30 \pm 0.06$ per 1000 $\mu m^2$; p=0.13; Student's *t*-test) (OSVZ: Control, $0.019 \pm 0.005$ per 1000 $\mu m^2$; sFGFR3, $0.013 \pm 0.010$ per 1000 $\mu m^2$; p=0.25; Student's *t*-test) (*Figure 3—figure supplement 2G,H*, *Supplementary file 1*), suggesting that FGF signaling is irrelevant to apoptosis.

## Upper layers are predominantly affected by FGF signaling

We examined whether the layer structure of the cortex was affected by sFGFR3 expression. Immunohistochemistry for FoxP2 and Ctip2, which are expressed in layers 5 and 6 (*Figure 5A,B*), demonstrated normal layer structure in the cerebral cortex electroporated with sFGFR3. Consistently, in situ hybridization for *Rorb* and *Cux1*, which are expressed in layer 4 and layers 2–4, respectively, showed the layer structure of the cerebral cortex was not affected by expressing sFGFR3 (*Figure 5C,D*). The locations of GFP-positive cells were not affected by sFGFR3; GFP-positive cells were distributed in layers 4 and 5 in both control and sFGFR3-transfected cortex (*Figure 5—figure supplement 1*). In addition, the locations of GFP-positive migrating cells and layer formation of the cerebral cortex during development seemed to be unaffected by sFGFR3 (*Figure 5—figure supplement 2*).

Finally, we quantified the thickness of cortical layers. Because it has been hypothesized that evolutional expansion of upper layers in gyrencephalic mammals resulted from an increase in the number of oRG cells (*Borrell and Götz, 2014*; *Lui et al., 2011*; *Lukaszewicz et al., 2005*; *Nowakowski et al., 2016*), it seemed plausible that the reduction of Pax6-positive/Tbr2-negative oRG cells by sFGFR3 led to preferential reduction of upper layers. Consistently, we found that the thickness of layer 2/3 was significantly reduced by sFGFR3 (Control, $0.97 \pm 0.06$; sFGFR3, $0.77 \pm 0.06$; p=0.017; Student's *t*-test) (*Figure 5E*, *Supplementary file 1*). In contrast, the thicknesses of layer 4, layer 5 and layer 6 were not affected by sFGFR3 (Layer 4: Control, $0.99 \pm 0.05$; sFGFR3, $0.96 \pm 0.06$; p=0.34; Student's *t*-test) (Layer 5: Control, $0.95 \pm 0.03$; sFGFR3, $1.12 \pm 0.13$; p=0.078; Student's *t*-test) (Layer 6: Control, $1.08 \pm 0.13$; sFGFR3, $1.00 \pm 0.08$; p=0.26; Student's *t*-test) (*Figure 5F–H*, *Supplementary file 1*). It is worth noting that GFP-positive transfected neurons were distributed in layers 4 and 5, but there were almost no GFP-positive cells in layer 2/3 (*Figure 5—figure supplement 1*), supporting our results that sFGFR3 exhibits a non-cell-autonomous effect. These findings suggest that FGF signaling affects the ratio between upper and lower layers of the cerebral cortex, and that the increased upper/lower ratio results in cortical folding. The most conceivable scenario would be that FGF increases OSVZ progenitors and preferentially increases upper layer neurons, generating the protrusion of cortical folds (*Figure 5I*).

## Discussion

Although the mechanisms underlying the formation of cortical folds have been of great interest, they had remained largely unknown. Here we have shown that regional differences in FGFR expression exist in the ferret cerebral cortex even before cortical folds are formed. Using our IUE technique, we uncovered that FGF signaling is required for cortical folding and the proliferation of OSVZ progenitors. Furthermore, FGF signaling preferentially affects the thickness of upper layers. We previously demonstrated that FGF overexpression was sufficient to increase cortical folds and OSVZ progenitors (*Masuda et al., 2015*). Furthermore, FGF overexpression preferentially increased the

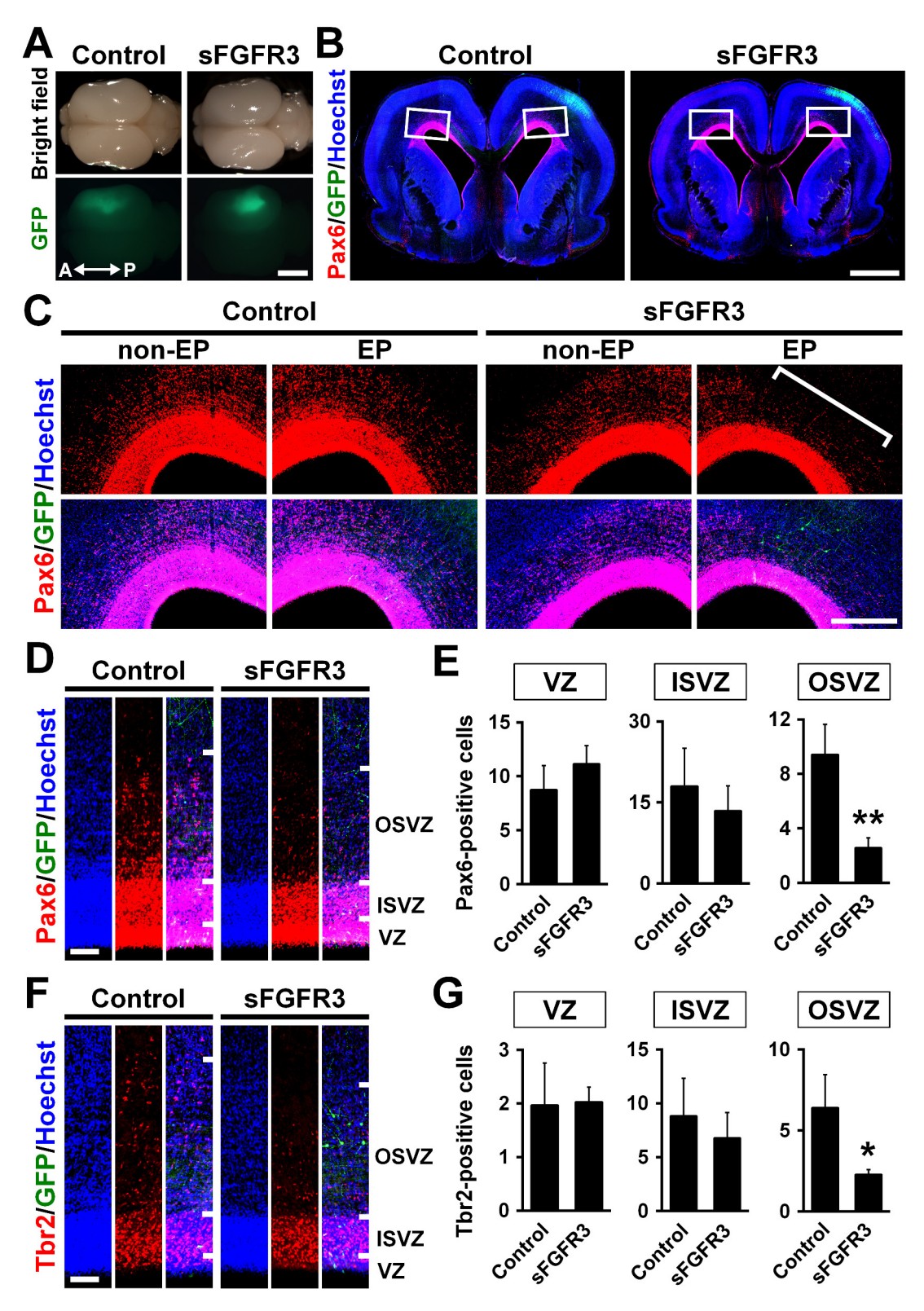

**Figure 3.** The roles of FGF signaling in neural progenitors of the developing ferret cortex. pCAG-EGFP plus either pCAG-sFGFR3 or pCAG control vector was electroporated at E33, and the brains were dissected at P6. (**A**) Dorsal views of the transfected brains. A, anterior; P, posterior. (**B**) Coronal sections stained with anti-Pax6 antibody and Hoechst 33342. Transfected regions were identified with GFP fluorescence. (**C**) Magnified images within the boxes of (**B**). Note that Pax6-positive cells were markedly decreased in the OSVZ underneath the sFGFR3-transfected region (square bracket). (**D**)
*Figure 3 continued on next page*

*Figure 3 continued*

Pax6-positive cells in the VZ, the ISVZ and the OSVZ of the transfected brain. (**E**) Quantification of the numbers of Pax6-positive cells per 10-µm-wide strips of the VZ, ISVZ and OSVZ. Pax6-positive cells were significantly reduced by sFGFR3 in the OSVZ selectively. n = 3 animals for each condition. Bars represent mean ± SD. (**F**) Tbr2-positive cells in the VZ, the ISVZ and the OSVZ of the transfected brain. (**G**) Quantification of Tbr2-positive cells. Tbr2-positive cells were significantly reduced by sFGFR3 in the OSVZ selectively. n = 3 animals for each condition. Bars present mean ± SD. *p<0.05, **p<0.01, Student's *t*-test. Scale bars = 5 mm (**A**), 2 mm (**B**), 500 µm (**C**) and 100 µm (**D and F**).

DOI: https://doi.org/10.7554/eLife.29285.012

The following figure supplements are available for figure 3:

**Figure supplement 1.** sFGFR reduced Pax6-positive/Tbr2-negative oRG cells and Pax6-positive/Tbr2-positive IP cells in the OSVZ.

DOI: https://doi.org/10.7554/eLife.29285.013

**Figure supplement 2.** FGF signaling is required for cell proliferation in the OSVZ of the developing ferret cortex.

DOI: https://doi.org/10.7554/eLife.29285.014

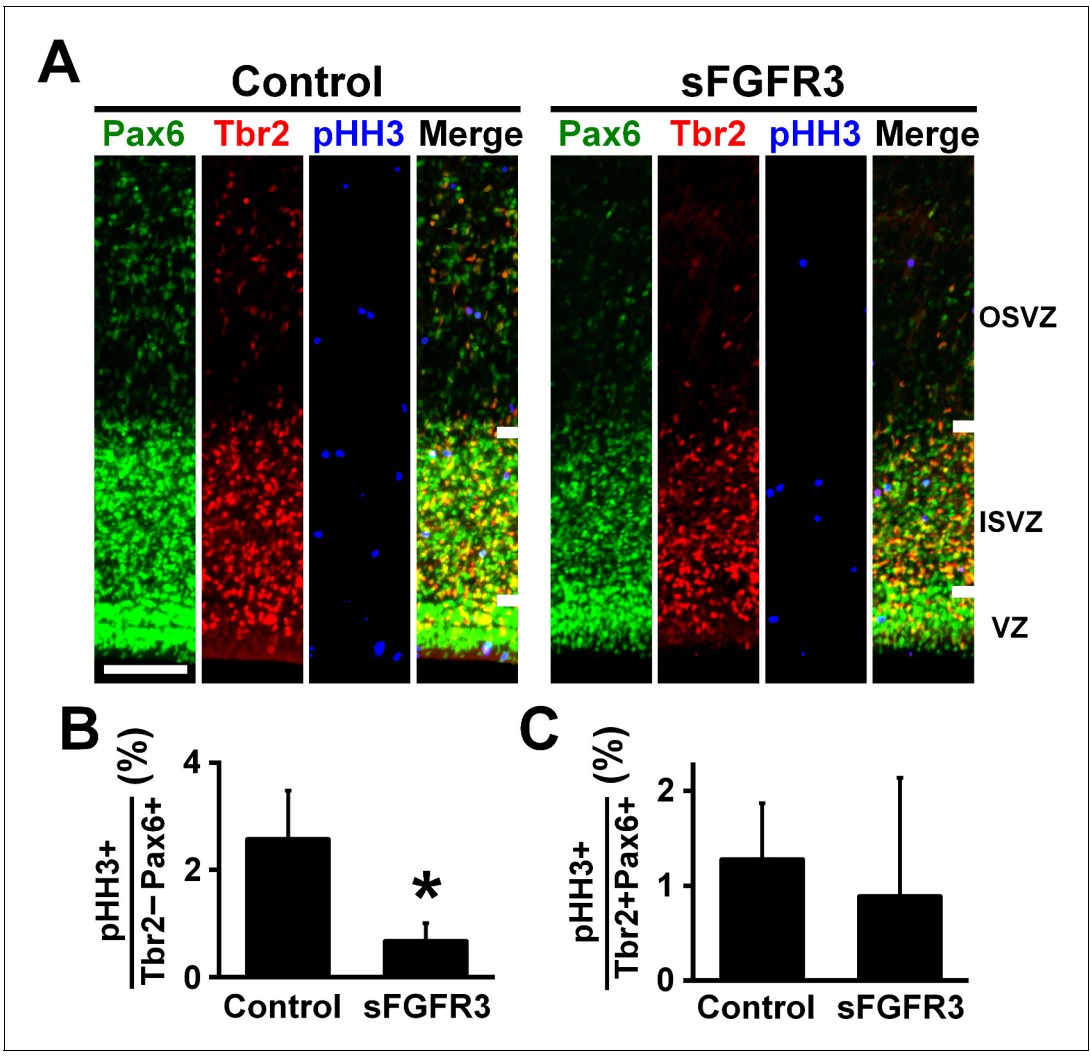

**Figure 4.** FGF signaling is required for cell proliferation of Pax6-positive, Tbr2-negative oRG cells. pCAG-EGFP plus either pCAG-sFGFR3 or pCAG control vector were electroporated at E33, and the brains were dissected at P6. (**A**) Coronal sections were triple-stained with anti-Pax6, anti-Tbr2 and anti-phospho-histone H3 (pHH3) antibodies. The VZ, the ISVZ and the OSVZ are shown. Scale bars = 100 µm. (**B**) The percentage of Tbr2-negative/Pax6-positive cells co-expressing pHH3. (**C**) The percentage of Tbr2-positive/Pax6-positive cells co-expressing pHH3. n = 3 animals for each condition. Bars present mean ± SD. *p<0.05, Student's *t*-test.

DOI: https://doi.org/10.7554/eLife.29285.015

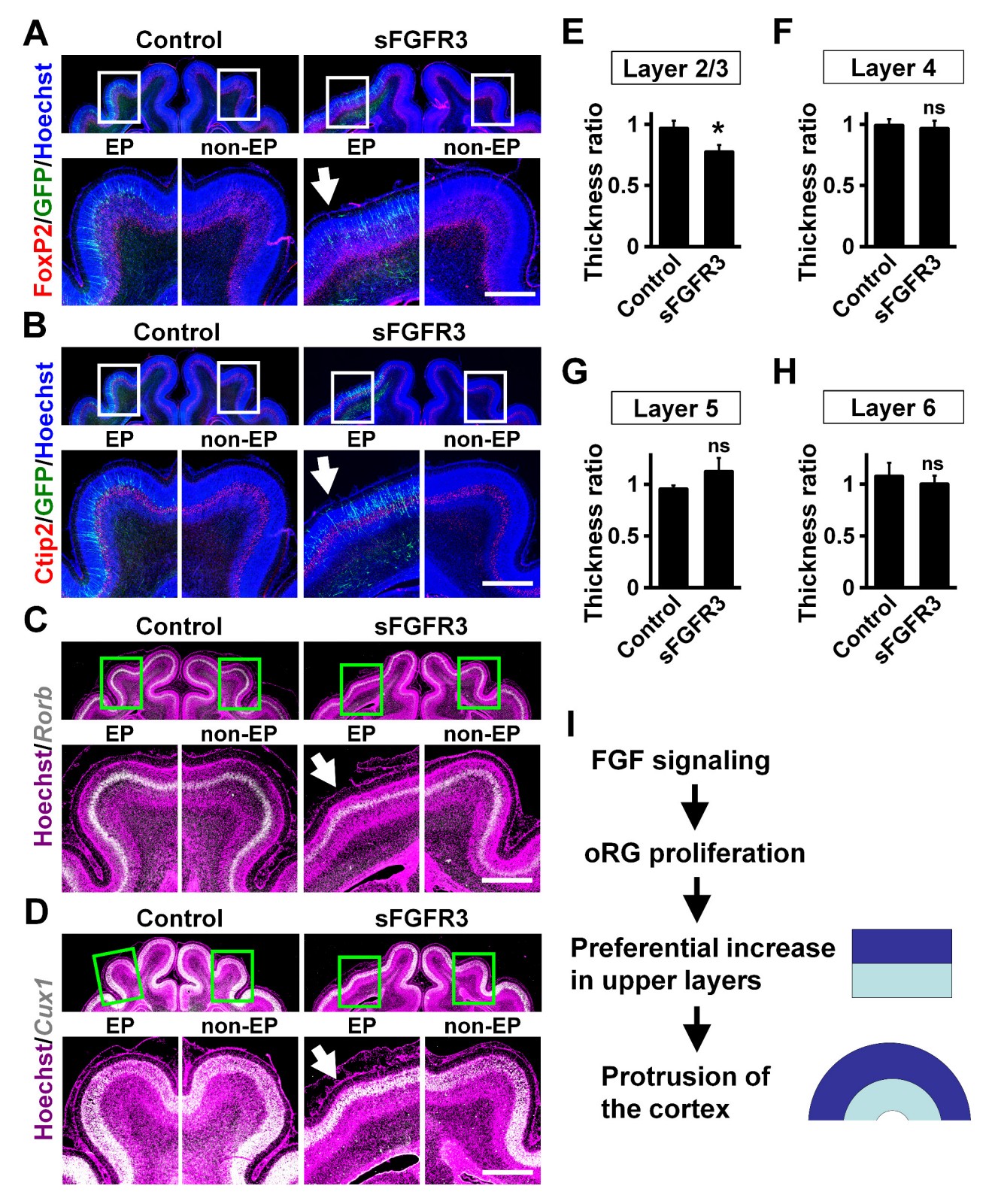

**Figure 5.** Upper layers are predominantly affected by inhibiting FGF signaling. pCAG-EGFP plus either pCAG-sFGFR3 or pCAG control vector was electroporated at E33, and the brains were dissected at P16. Coronal sections were subjected to Hoechst 33342 staining plus either immunohistochemistry or in situ hybridization. (**A**) FoxP2 immunohistochemistry. (**B**) Ctip2 immunohistochemistry. (**C**) *Rorb* in situ hybridization (white). (**D**) *Cux1* in situ hybridization (white). The images within the boxes in the upper panels were magnified and are shown in the lower panels. Note that

*Figure 5 continued on next page*

*Figure 5 continued*

sFGFR3 did not affect the layer structure of the cerebral cortex while cortical folding was inhibited (sFGFR3, EP, arrows). Scale bars = 1 mm. (E–H) Quantification of the thicknesses of layer 2/3 (E), layer 4 (F), layer 5 (G) and layer 6 (H). The ratios of the thickness of the electroporated side relative to that of the non-electroporated side are shown. Note that the thickness ratio of layer 2/3 was significantly reduced by sFGFR3, while those of layer 4, layer 5 and layer 6 were not. n = 3 animals for each condition. Bars present mean ± SD. *p<0.05; ns, not significant; Student's *t*-test. (I) A model of the mechanisms underlying the formation of cortical folds in the gyrencephalic brain.

DOI: https://doi.org/10.7554/eLife.29285.016

The following figure supplements are available for figure 5:

**Figure supplement 1.** Distribution of GFP-positive cells in the sFGFR3-transfected cerebral cortex.

DOI: https://doi.org/10.7554/eLife.29285.017

**Figure supplement 2.** Layer structures of the cerebral cortex and laminar positions of GFP-positive cells in the sFGFR3-transfected brains during development.

DOI: https://doi.org/10.7554/eLife.29285.018

thickness of layer 2/3 (*Masuda et al., 2015*). Taken together, our findings indicate that FGF signaling is crucial for cortical folding in gyrencephalic mammals and is a pivotal upstream regulator of the production of OSVZ progenitors. FGF signaling is the first signaling pathway found to regulate cortical folding.

## Involvement of SVZ progenitors in cortical folding

Our results showed that sFGFR3 inhibited both cortical folding and the production of SVZ progenitors. Our previous study showed that activation of FGF signaling resulted in both additional cortical folds and increased SVZ progenitors (*Masuda et al., 2015*). These correlations between cortical folding and SVZ progenitors raised the possibility that SVZ progenitors are responsible for generating cortical folds. Consistently, previous pioneering studies using monkeys reported that even before cortical folds are formed during development, SVZ progenitors are abundant in the cortical regions where the gyri will be formed (*Smart et al., 2002*). Recently, we directly tested this possibility and uncovered that the formation of cortical folds was inhibited when SVZ progenitors were reduced by expressing dominant-negative Tbr2 transcription factor (DN-Tbr2) (*Toda et al., 2016*). Thus, these results suggest that SVZ progenitors are indeed crucial for cortical folding during development.

It has been hypothesized that the ratio between upper layers and lower layers of the cerebral cortex is crucial for cortical folding (*Richman et al., 1975*; *Kriegstein et al., 2006*). A preferential increase in upper layers relative to lower layers is supposed to produce the protrusion (i.e. the gyrus) of the cortex. Consistent with this hypothesis, current and previous our findings demonstrated that sFGFR3 and DN-Tbr2 preferentially reduced upper layers where cortical folding was impaired (*Toda et al., 2016*). Furthermore, when FGF signaling was enhanced, upper layers were predominantly increased and additional gyri were formed (*Masuda et al., 2015*).

Besides the ratio between upper layers and lower layers mentioned above, several hypotheses have been proposed to explain the mechanisms of cortical folding (*Lewitus et al., 2013*; *Sun and Hevner, 2014*). It was previously proposed that radial fibers provided by oRG cells are involved in gyrification (*Reillo et al., 2011*). In prospective gyral regions, oRG cells provide oblique radial fibers for conical neuronal migration, which could contribute to the expansion of the cortical surface. Because our results showed that the numbers of oRG cells were reduced by sFGFR3, reduction of oRG fibers could also be involved in the effect of sFGFR3. Another hypothesis is that the tension of nerve fibers plays a key role in gyrification (*Van Essen, 1997*). This also seemed possible because axonal trajectories could also be affected by sFGFR3. Future investigations would be necessary in order to gain a complete understanding of the mechanisms underlying cortical folding.

## The roles of FGF signaling in cortical folding and the production of SVZ progenitors

Our findings demonstrated that the protrusion of the gyrus was reduced by sFGFR3, whereas the pattern of the gyrus was preserved. These results suggest that FGF signaling is involved in making the protrusion of the gyri rather than regulating the positions of the gyri. Consistently, the overall gyral patterns seemed to be unaffected by FGF8, although small additional gyri appeared (*Masuda et al., 2015*). On the other hand, the role of FGF signaling in determining the number of

the gyri is unclear. sFGFR3 did not seem to affect the number of the gyri, though FGF8 increased it, leading to polymicrogyria (*Masuda et al., 2015*). Because a previous study reported that an increase in the number of neurons resulted in additional folding (*Nonaka-Kinoshita et al., 2013*), the increase in neurons induced by FGF8 seemed to be sufficient for producing additional gyri.

Our findings indicate that FGF signaling regulates cortical folding and the proliferation of OSVZ progenitors. In contrast to OSVZ progenitors, the proliferation of RG cells in the VZ was not affected by inhibition of FGF signaling, even though FGFR is expressed in the VZ. Previous studies have demonstrated that several ligands (e.g. growth factors and morphogens) other than FGF, such as BMP7, promoted the proliferation of RG cells (*Segklia et al., 2012*). Therefore, it seemed plausible that FGFs and other growth factors act redundantly to promote the proliferation of RG cells in the VZ. It should be noted that although our findings indicate that FGF signaling regulates the proliferation of OSVZ progenitors, it remains possible that FGF also regulates the differentiation of RG cells into SVZ progenitors because FGFRs are also expressed in RG cells. Future investigations would be necessary for addressing these points.

Although our findings clearly uncovered a crucial role of FGF signaling in cortical folding, currently it is unknown which types of FGFR and FGF are responsible for cortical folding. Our in situ hybridization showed that FGFR1, 2 and 3 were expressed in the developing ferret cortex. Because similar downstream signaling pathways are activated by FGFR1, 2 and 3 (*Iwata and Hevner, 2009*), it seems likely that they work redundantly. Similarly, because several FGFs are expressed in the developing mouse cortex (Allen Brain Atlas), they may work cooperatively.

## Advantages of ferrets for examining the mechanisms of cortical folding

Although several hypotheses about the mechanisms of cortical folding have been proposed (*Kelava et al., 2013*; *Fernández et al., 2016*), it had been difficult to test these hypotheses directly through experiments. This was mainly because efficient genetic manipulation techniques that can be used for gyrencephalic mammals were missing. Taking advantage of our IUE technique for ferrets, here we uncovered the crucial role of FGF signaling in cortical folding. Using the same technique, we previously reported that SVZ progenitors were indispensable for cortical folding (*Toda et al., 2016*), and produced polymicrogyria model ferrets (*Masuda et al., 2015*). Ferrets are clearly an important option for investigating the mechanisms of cortical folding.

Ferrets have several important advantages. First, anatomical and electrophysiological information about the ferret brain is available because the ferret brain has been widely used for anatomical and electrophysiological research. The structures of cortical gyri and sulci are well described (*Smart and McSherry, 1986b*). Second, the processes of the formation of cortical gyri during development have been reported (*Smart and McSherry, 1986b*; *Smart and McSherry, 1986a*; *Neal et al., 2007*). For example, when ferret babies are born, their brains are lissencephalic, and cortical folds are formed after birth. Therefore, the mechanisms of cortical folding can be analyzed using neonatal ferrets rather than embryos. Third, usually more than six ferret babies are born from one pregnant mother. This large number of babies per pregnant mother facilitates examination under various experimental conditions and the ability to obtain a sufficient number of experimental samples. Finally, because our IUE technique is now available, genetically manipulated ferret brains can be obtained within a few weeks. Uncovering the molecular mechanisms underlying the formation of gyrencephalic brains using ferrets would help lead to our ultimate goal of understanding the human brain and its diseases.

## Materials and methods

**Key resources table**

| Reagent type (species) or resource | Designation | Source or reference | Identifiers | Additional information |
|---|---|---|---|---|
| antibody | anti-Tbr2 (rabbit polyclonal) | Abcam | Cat# ab23345; RRID: AB_778267 | N/A |
| antibody | anti-Pax6 (mouse monoclonal) | Abcam | Cat# ab78545; RRID: AB_1566562 | N/A |

*Continued on next page*

*Continued*

| Reagent type (species) or resource | Designation | Source or reference | Identifiers | Additional information |
|---|---|---|---|---|
| antibody | anti-Pax6 (rabbit polyclonal) | Covance | Cat# PRB-278P; RRID: AB_291612 | N/A |
| antibody | anti-Ki-67 (rabbit polyclonal) | Leica | Cat# NCL-Ki67p; RRID: AB_442102 | N/A |
| antibody | anti-phospho-histone H3 (mouse monoclonal) | Millipore | Cat# 05–806; RRID: AB_310016 | N/A |
| antibody | anti-phosphorylated vimentin (mouse monoclonal) | Medical and Biological Laboratories, Japan | Cat# D076-3; RRID: AB_592963 | N/A |
| antibody | anti-cleaved caspase 3 (rabbit monoclonal) | BD Pharmingen | Cat# 559565; RRID: AB_397274 | N/A |
| antibody | anti-Ctip2 (rat monoclonal) | Abcam | Cat# ab18465; RRID: AB_2064130 | N/A |
| antibody | anti-FOXP2 (rabbit polyclonal) | Atlas antibodies | Cat# HPA000382; RRID: AB_1078908 | N/A |
| antibody | anti-GFAP (mouse monoclonal) | Sigma-Aldrich | Cat# G3893; RRID: AB_477010 | N/A |
| antibody | anti-GFP (rat monoclonal) | Nacalai tesque, Japan | Cat# 440426; RRID: AB_2313652 | N/A |
| antibody | anti-GFP (rabbit polyclonal) | Medical and Biological Laboratories, Japan | Cat# 598; RRID: AB_591819 | N/A |
| antibody | biotin-conjugated anti-phospho-histone H3 | Millipore | Cat# 16–189; RRID: AB_310794 | N/A |
| antibody | alkaline phosphatase-conjugated anti-digoxigenin | Roche, Indianapolis, IN | Cat# 11093274910; RRID: AB_514497 | N/A |
| sequence-based reagent (*Mustela putorius furo*) | Ferret *FGFR1* forward1 (primer) | this paper | N/A | GGAGCTGGAAGTGCCTCCTCTTCTG |
| sequence-based reagent (*Mustela putorius furo*) | Ferret *FGFR1* reverse1 (primer) | this paper | N/A | TGATGCGGGTACGGTTGCTT |
| sequence-based reagent (*Mustela putorius furo*) | Ferret *FGFR1* forward2 (primer) | this paper | N/A | CAGGGGAGGAGGTGGAGGTG |
| sequence-based reagent (*Mustela putorius furo*) | Ferret *FGFR1* reverse2 (primer) | this paper | N/A | CGGCACCGCATGCAATTTCTTTTCCATC |
| sequence-based reagent (*Mustela putorius furo*) | Ferret *Sprouty2* forward (primer) | this paper | N/A | ATCGCAGGAAGACGAGAATCCAAGG |
| sequence-based reagent (*Mustela putorius furo*) | Ferret *Sprouty2* reverse (primer) | this paper | N/A | CTGGGTGGGACAGTGGGAACTTTGC |
| sequence-based reagent (*Mus musculus*) | Mouse *Fgfr1* forward (primer) | this paper | N/A | CTGCATGGTTGACCGTTCTGGAAGC |
| sequence-based reagent (*Mus musculus*) | Mouse *Fgfr1* reverse (primer) | this paper | N/A | TGTAGATCCGGTCAAACAACGCCTC |
| sequence-based reagent (*Mus musculus*) | Mouse *Fgfr2* forward (primer) | this paper | N/A | GGAAGGAGTTTAAGCAGGAGCATCG |
| sequence-based reagent (*Mus musculus*) | Mouse *Fgfr2* reverse (primer) | this paper | N/A | CGATTCCCACTGCTTCAGCCATGAC |
| sequence-based reagent (*Mus musculus*) | Mouse *Fgfr3* forward (primer) | this paper | N/A | GAAAGTGTGGTACCCTCCGATCGTG |
| sequence-based reagent (*Mus musculus*) | Mouse *Fgfr3* reverse (primer) | this paper | N/A | GTCCAAAGCAGCCTTCTCCAAGAGG |
| sequence-based reagent (*Mustela putorius furo*) | Ferret *FGFR2* forward (primer) | this paper | N/A | AGAGATAAGCTGACGCTGGGCAAACC |
| sequence-based reagent (*Mustela putorius furo*) | Ferret *FGFR2* reverse (primer) | this paper | N/A | GAGGAAGGCAGGGTTCGTAAGGC |

*Continued on next page*

*Continued*

| Reagent type (species) or resource | Designation | Source or reference | Identifiers | Additional information |
|---|---|---|---|---|
| sequence-based reagent (*Mustela putorius furo*) | Ferret *FGFR3* forward (primer) | this paper | N/A | GAGGCTAAATTACGGGTACCTGA |
| sequence-based reagent (*Mustela putorius furo*) | Ferret *FGFR3* reverse (primer) | this paper | N/A | GAGAACAAAGACCACCCTGAAC |
| recombinant DNA reagent | pCAG-sFGFR3 (plasmid) | PMID: 11567107 | N/A | N/A |
| recombinant DNA reagent | pCAG-EGFP (plasmid) | PMID: 20181605 | N/A | N/A |
| recombinant DNA reagent | pCAG control (plasmid) | PMID: 26482531 | N/A | N/A |
| recombinant DNA reagent (*Mustela putorius furo*) | pCRII-ferret *FGFR1*_1 (plasmid) | this paper | N/A | vector: pCRII; cDNA fragment: ferret *FGFR1*. |
| recombinant DNA reagent (*Mustela putorius furo*) | pCRII-ferret *FGFR1*_2 (plasmid) | this paper | N/A | vector: pCRII; cDNA fragment: ferret *FGFR1*. |
| recombinant DNA reagent (*Mustela putorius furo*) | pCRII-ferret *FGFR2* (plasmid) | PMID: 26482531 | N/A | vector: pCRII; cDNA fragment: ferret *FGFR2*. |
| recombinant DNA reagent (*Mustela putorius furo*) | pCRII-ferret *FGFR3* (plasmid) | PMID: 26482531 | N/A | vector: pCRII; cDNA fragment: ferret *FGFR3*. |
| recombinant DNA reagent (*Mustela putorius furo*) | pCRII-ferret *Sprouty2* (plasmid) | this paper | N/A | vector: pCRII; cDNA fragment: ferret *Sprouty2*. |
| recombinant DNA reagent (*Mus musculus*) | pCRII-mouse *Fgfr1* (plasmid) | this paper | N/A | vector: pCRII; cDNA fragment: mouse *Fgfr1*. |
| recombinant DNA reagent (*Mus musculus*) | pCRII-mouse *Fgfr2* (plasmid) | this paper | N/A | vector: pCRII; cDNA fragment: mouse *Fgfr2*. |
| recombinant DNA reagent (*Mus musculus*) | pCRII-mouse *Fgfr3* (plasmid) | this paper | N/A | vector: pCRII;cDNA fragment: mouse *Fgfr3*. |
| recombinant DNA reagent (*Mustela putorius furo*) | pCRII-Ferret *Cux1* (plasmid) | PMID: 20575059 | N/A | N/A |
| recombinant DNA reagent (*Mustela putorius furo*) | pCRII-Ferret *Rorb* (plasmid) | PMID: 20575059 | N/A | N/A |
| software, algorithm | FIJI | http://fiji.sc | RRID:SCR_002285 | N/A |

## Animals

Normally pigmented, sable ferrets (*Mustela putorius furo*) were purchased from Marshall Farms (North Rose, NY). Ferrets were maintained as described previously (*Kawasaki et al., 2004*; *Iwai and Kawasaki, 2009*; *Iwai et al., 2013*). ICR mice were purchased from SLC (Hamamatsu, Japan) and were reared on a normal 12 hr light/dark schedule. The day of conception and that of birth were counted as embryonic day 0 (E0) and postnatal day 0 (P0), respectively. All procedures were approved by the Animal Care Committee of Kanazawa University. Experiments were repeated at least three times and gave consistent results.

## *In utero* electroporation (IUE) procedure for ferrets

The IUE procedure to express transgenes in the ferret brain was described previously (*Kawasaki et al., 2012*, *2013*). Briefly, pregnant ferrets at E33 were anesthetized, and their body temperature was monitored and maintained using a heating pad. The uterine horns were exposed and kept wet by adding drops of PBS intermittently. The location of embryos was visualized with transmitted light delivered through an optical fiber cable. The pigmented iris was visible, and this enabled us to assume the location of the lateral ventricle. Approximately 2–5 µl of DNA solution was injected into the lateral ventricle at the indicated ages using a pulled glass micropipette. Each embryo within the uterus was placed between tweezer-type electrodes with a diameter of 5 mm (CUY650-P5; NEPA Gene, Japan). Square electric pulses (50–100 V, 50 ms) were passed 5 times at 1 s intervals using an electroporator (ECM830, BTX, Holliston, MA). The wall and skin of the abdominal cavity were sutured, and the embryos were allowed to develop normally.

## Plasmids

pCAG-EGFP was described previously (*Sehara et al., 2010*). pCAG-sFGFR3 was made in accordance with a previous report (*Fukuchi-Shimogori and Grove, 2001*). Briefly, the extracellular domain of human FGFR3c was amplified using the following primers and was subcloned into pCAG plasmid vector, yielding pCAG-sFGFR3. Forward, CCATGGGCGCCCCTGCC; reverse, GGAGCCCAGGCC TTTCTT. Plasmids were purified using the Endofree Plasmid Maxi Kit (Qiagen, Valencia, CA). For co-transfection, a mixture of pCAG-EGFP plus either pCAG-sFGFR3 or pCAG control plasmid was used. Prior to IUE procedures, plasmid DNA was diluted to 2.5–6.0 mg/ml in PBS, and Fast Green solution was added to a final concentration of 0.5% to monitor the injection.

## Preparation of sections

Preparation of sections was performed as described previously with slight modifications (*Toda et al., 2013*; *Kawasaki et al., 2000*). Briefly, ferrets or mice were deeply anesthetized with pentobarbital and transcardially perfused with 4% paraformaldehyde (PFA), and the brains were dissected. Then, the brains were cryoprotected by three-days immersion in 30% sucrose and embedded in OCT compound. Sections of 18 or 50 μm thickness were prepared using a cryostat.

## Microscopy

Epifluorescence microscopy and confocal microscopy were performed with a BIOREVO BZ-9000 (Keyence, Japan) and a FLUOVIEW FV10i (Olympus, Japan), respectively.

## Immunohistochemistry

Immunohistochemistry was performed as described previously with slight modifications (*Toda et al., 2013*; *Kawasaki et al., 2000*). Coronal sections were permeabilized with 0.3% Triton X-100/PBS and incubated overnight with primary antibodies, which included anti-Tbr2 (Abcam, UK, RRID: AB_778267), anti-Pax6 (Covance, Princeton, NJ, RRID: AB_291612), anti-Ki-67 (Leica, Germany, RRID: AB_442102), anti-phospho-histone H3 (Millipore, Billerica, MA, RRID: AB_310016), anti-phosphorylated vimentin (Medical and Biological Laboratories, Japan, RRID: AB_592963), anti-cleaved caspase 3 (BD Pharmingen, San Diego, CA, RRID: AB_397274), anti-Ctip2 (Abcam, UK, RRID: AB_2064130), anti-FOXP2 (Atlas antibodies, Sweden, RRID: AB_1078908), anti-GFAP (Sigma-Aldrich, St. Louis, MO, RRID: AB_477010) and anti-GFP antibodies (Nacalai tesque, Japan, RRID: AB_2313652; Medical and Biological Laboratories, Japan, RRID: AB_591819). After incubation with secondary antibodies and Hoechst 33342, the sections were washed and mounted.

For triple immunostaining, after double immunostaining was performed as described above, the sections were incubated with biotin-conjugated anti-phospho-histone H3 antibody (Millipore, Billerica, MA, RRID: AB_310794) and subsequently with fluorescent-dye conjugated streptavidin.

## In situ hybridization

In situ hybridization was performed as described previously (*Matsumoto et al., 2017*). Sections prepared from fresh-frozen or fixed tissues were treated with 4% PFA for 10 min, 1 μg/ml proteinase K for 10 min and 0.25% acetic anhydride for 10 min. After prehybridization, the sections were incubated overnight at 58 ˚C with digoxigenin-labeled RNA probes diluted in hybridization buffer (50% formamide, 5x SSC, 5x Denhardt's solution, 0.3 mg/ml yeast RNA, 0.1 mg/ml herring sperm DNA, and 1 mM DTT). The sections were then incubated with alkaline phosphatase-conjugated anti-digoxigenin antibody (Roche, Indianapolis, IN, RRID: AB_514497) and Hoechst 33342, and were visualized using NBT/BCIP as substrates.

For combination of in situ hybridization and immunostaining, after hybridization was performed, the sections were incubated with anti-Pax6 (Abcam, UK, RRID: AB_1566562), anti-Tbr2, anti-GFP (Medical and Biological Laboratories, Japan, RRID: AB_591819) and alkaline phosphatase-conjugated anti-digoxigenin antibodies. After being incubated with secondary antibodies, in situ signals were visualized with NBT/BCIP. Probes used here were as follows. Ferret *FGFR2* and *FGFR3* probes were described previously (*Masuda et al., 2015*). Two ferret FGFR1 cDNA fragments were amplified by RT-PCR and inserted into the pCRII vector. The sequences of primers used to amplify ferret *FGFR1* cDNA fragments were as follows: forward1, GGAGCTGGAAGTGCCTCCTCTTCTG; reverse1,

TGATGCGGGTACGGTTGCTT; forward2, CAGGGGAGGAGGTGGAGGTG; reverse2 CGGCACCGCATGCAATTTCTTTTCCATC. A mixture of two probes made from these two *FGFR1* cDNA fragments were used as the *FGFR1* probe. Ferret *Sprouty2,* mouse *Fgfr1,* mouse *Fgfr2 and* mouse *Fgfr3* cDNA fragments were amplified using RT-PCR and inserted into the pCRII vector. The sequences of primers used to amplify ferret *Sprouty2,* mouse *Fgfr1,* mouse *Fgfr2 and* mouse *Fgfr3* cDNA fragments were as follows: ferret *Sprouty2,* forward ATCGCAGGAAGACGAGAATCCAAGG, reverse CTGGGTGGGACAGTGGGAACTTTGC; mouse *Fgfr1,* forward CTGCATGGTTGACCGTTC TGGAAGC, reverse TGTAGATCCGGTCAAACAACGCCTC; mouse *Fgfr2,* forward GGAAGGAG TTTAAGCAGGAGCATCG, reverse CGATTCCCACTGCTTCAGCCATGAC; mouse *Fgfr3,* forward GAAAGTGTGGTACCCTCCGATCGTG, reverse GTCCAAAGCAGCCTTCTCCAAGAGG. The ferret *Cux1* and *Rorb* probes were described previously (*Matsumoto et al., 2017*; *Rowell et al., 2010*; *Shinmyo et al., 2017*).

## RT-PCR

RT-PCR was performed as described previously with modification (*Kawasaki et al., 2002*). Total RNA was isolated from the P0 ferret cerebral cortex using the RNeasy Mini Kit (Qiagen, Valencia, CA). Reverse transcription was performed using oligo(dT)12–18 (Thermo Fisher Scientific, Waltham, MA) and Superscript III (Thermo Fisher Scientific, Waltham, MA). Samples without Superscript III were also made as negative controls. PCR reaction was performed using the following primers: *FGFR1*-forward, GGAGCTGGAAGTGCCTCCTCTTCTG; *FGFR1*-reverse, CGGCACCGCATGCAA TTTCTTTTCCATC; *FGFR2*-forward, AGAGATAAGCTGACGCTGGGCAAACC; *FGFR2*-reverse, GAG-GAAGGCAGGGTTCGTAAGGC; *FGFR3*-forward, GAGGCTAAATTACGGGTACCTGA; *FGFR3*-reverse, GAGAACAAAGACCACCCTGAAC. PCR products made using these primers were confirmed by DNA sequencing.

## Cell counting

Coronal sections were stained with anti-Pax6, anti-Tbr2, anti-Ki-67, anti-pHH3, anti-pVim and anti-cleaved caspase 3 antibodies and Hoechst 33342. After background signals were removed using the 'threshold' tool of ImageJ software (Default or MaxEntropy option), the numbers of immunopositive cells in GFP-positive areas were manually counted using the 'cell counter' tool and were divided by the total area of the ROI to calculate cell densities. To calculate the numbers of Pax6-positive and Tbr2-positive cells, coronal sections containing the striatum were used. The densities of Pax6-positive and Tbr2-positive cells in each germinal layer in dorsolateral regions of the cerebral cortex were multiplied by its thickness. One region per each animal was used. The thickness of each germinal layer was measured as follows using three sections for each animal: the area of each germinal layer was divided by its tangential length. The cell-dense layer next to the VZ was identified as the ISVZ and the cell-sparse layer between the ISVZ and the IZ was identified as the OSVZ. In addition, the border between the ISVZ and the OSVZ was also identified by the presence of GFP-positive fibers (*Kawasaki et al., 2013*).

## Calculation of the local GS ratio, the local SD ratio and the local GI ratio

Serial coronal sections containing the suprasylvian gyrus (SSG) were prepared from electroporated brains. One section per every 100 µm was selected. The selected three serial sections containing the SSG for each brain were stained with Hoechst 33342, and images of whole sections were acquired using a BZ-9000 microscope (Keyence, Japan). The averages of the local GS, local SD and local GI ratios were calculated using the three sections for each animal.

To calculate the local GS ratio, the area surrounded by the brain surface (*Figure 1—figure supplement 2*, green line) and the red line connecting the bottom of the suprasylvian sulcus (SSS) and that of the lateral sulcus (LS) was measured (local GS value) (*Figure 1—figure supplement 2*). To minimize the variation of the local GS values depending on the positions of coronal sections in the brain, the local GS value on the electroporated side and that on the contralateral non-electroporated side of the cerebral cortex in the same brain sections were measured, and the former was divided by the latter (local GS ratio). The local GS ratio would be 1 if the size of SSG was the same between the electroporated side and the other side (i.e. non-electroporated side), and would be smaller than 1 if cortical folding was suppressed by genetic manipulation.

To calculate the local SD ratio, a line connecting the top of the SSG and the top of the middle ectosylvian gyrus (MEG) was drawn (*Figure 1—figure supplement 3*, red line). Then a green line connecting the bottom of the suprasylvian sulcus (SSS) and the red line was drawn, so that the green line was perpendicular to the red line (*Figure 1—figure supplement 3*). The length of the green line was used as the local SD value. To minimize the variation of the local SD values depending on the positions of coronal sections in the brain, the local SD value on the electroporated side and that on the contralateral non-electroporated side of the cerebral cortex in the same brain sections were measured, and the former was divided by the latter (local SD ratio). The local SD ratio would be 1 if the depth of the SSS was the same between the electroporated side and the other side (i.e. non-electroporated side), and would be smaller than 1 if the depth of the SSS was reduced by genetic manipulation.

To calculate the local GI ratio, a line connecting the top of the SSG and that of the MEG was drawn (*Figure 1—figure supplement 4*, red line). The length of the complete contour between the top of the SSG and that of the MEG (*Figure 1—figure supplement 4*, green line) was then divided by that of the red line (local GI value). To minimize the variation of the local GI values depending on the positions of coronal sections in the brain, the local GI value on the electroporated side and that on the contralateral non-electroporated side of the cerebral cortex in the same brain sections were measured, and the former was divided by the latter (local GI ratio). The local GI ratio would be 1 if cortical folding was the same between the electroporated side and the other side (i.e. non-electroporated side), and would be smaller than 1 if the cortical folding was suppressed by genetic manipulation.

## Quantification of cortical thickness

Coronal sections containing the SSG were used for quantification. The sections were subjected to Hoechst 33342 staining plus either in situ hybridization for *Rorb* or immunohistochemistry for Ctip2, and images of whole sections were acquired using a BZ-9000 microscope. To calculate the thickness of each cortical layer in the SSG, the thicknesses of layer 2/3 and layer 4 in the SSG, which were identified using Hoechst and *Rorb* signals, and those of layer 5 and layer 6 in the SSG, which were identified using Hoechst and Ctip2 signals, were measured, and were subsequently divided by the tangential lengths of these areas. To minimize the variation of the thickness value depending on the positions of coronal sections in the brain, the thickness value on the electroporated side and that on the contralateral non-electroporated side of the cerebral cortex in the same brain sections were measured, and the former was divided by the latter (thickness ratio). The thickness ratio would be 1 if the thickness was the same between the electroporated side and the other side (i.e. non-electroporated side), and would be smaller than 1 if the thickness was reduced by genetic manipulation.

## Quantification of the distribution of *FGFR* signaling

Coronal sections prepared at E40 were subjected to in situ hybridization for *FGFR2* and *FGFR3* and Hoechst 33342 staining, and images of whole sections were acquired using a BZ-9000 microscope (Keyence, Japan). The borders of the OSVZ were determined using Hoechst images. After background signals were subtracted, total FGFR signal intensities per area were measured. The entire OSVZ regions were selected and straightened using the 'Straighten' function of ImageJ. Signal intensities along the tangential axis were then measured using the 'Plot Profile' of ImageJ.

## Acknowledgement

We apologize to all of the researchers whose work could not be cited due to space limitation. We are grateful to Drs. Eisuke Nishida (Kyoto University), the late Yoshiki Sasai and Shigetada Nakanishi (Suntory Foundation for Life Science) for their continuous encouragement. We thank Dr. Clifton W Ragsdale (University of Chicago) for plasmids, and Zachary Blalock, Dr. Tomohisa Toda, Dr. Yoshio Hoshiba, Miwako Hirota, Yuki Nishita and Kawasaki lab members for their helpful support. This work was supported by Grants-in-Aid for Scientific Research from the Ministry of Education, Culture, Sports, Science and Technology (MEXT), Life Science Foundation of Japan Senri Life Science Foundation, the Uehara Memorial Foundation and Takeda Science Foundation.

## Additional information

### Funding

| Funder | Author |
| --- | --- |
| Ministry of Education, Culture, Sports, Science, and Technology | Hiroshi Kawasaki |
| Senri Life Science Foundation | Hiroshi Kawasaki |
| Uehara Memorial Foundation | Hiroshi Kawasaki |
| Takeda Science Foundation | Hiroshi Kawasaki |

The funders had no role in study design, data collection and interpretation, or the decision to submit the work for publication.

### Author contributions

Naoyuki Matsumoto, Conceptualization, Resources, Data curation, Formal analysis, Validation, Investigation, Methodology, Writing—original draft, Writing—review and editing; Yohei Shinmyo, Formal analysis, Methodology; Yoshie Ichikawa, Data curation; Hiroshi Kawasaki, Conceptualization, Supervision, Funding acquisition, Writing—original draft, Writing—review and editing

### Author ORCIDs

Hiroshi Kawasaki (iD) http://orcid.org/0000-0002-2514-1497

### Ethics

Animal experimentation: All procedures were approved by the Animal Care Committee of Kanazawa University.

### Decision letter and Author response

Decision letter https://doi.org/10.7554/eLife.29285.021
Author response https://doi.org/10.7554/eLife.29285.022

## Additional files

### Supplementary files

• Supplementary file 1. Raw data for the quantification.
DOI: https://doi.org/10.7554/eLife.29285.019

• Transparent reporting form
DOI: https://doi.org/10.7554/eLife.29285.020

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
