## [Decision Letter]

Thank you for submitting your article "Gyrification of the cerebral cortex requires FGF signaling in the mammalian brain" for consideration by *eLife*. Your article has been reviewed by three peer reviewers, and the evaluation has been overseen by Joseph Gleason as the Reviewing Editor and Marianne Bronner as the Senior Editor. The following individual involved in review of your submission has agreed to reveal his identity: Edwin Monuki (Reviewer #1).

The reviewers have discussed the reviews with one another and the Reviewing Editor has drafted this decision to help you prepare a revised submission.

Summary:

Previous work by Masuda et al., 2015 using FGF8 in utero electroporation (IUE) showed surprising extra "normal-appearing" sulci and gyri, and previous work by the Borrell group to show differential expression between cortical gyri and sulci for FGFR3. The current sFGFR3 IUE work to decrease FGF signaling is a natural complement to the Masuda et al. study, and especially relevant because it was performed in the ferret neocortex, which maintains cardinal features of gyrate mammalian brain – i.e. folded pial surfaces paralleled by curved and continuous cortical layers, which generally overlie an unfolded ventricular surface. By performing an in utero knock-down of FGFR3, the authors have been able to demonstrate the necessity of this gene for cortical folding. Because the experiments are performed in a gyrencephalic species, this study provides the most clear insight into the molecular mechanisms of gyrification to date. The data is of high quality.

Essential revisions:

Given that the authors recognize non-cell autonomous role of FGFR3 in regulating FGF signaling,the authors should perform in situ hybridization for a few of the downstream effectors of FGF signaling and correlate the phenotype with in utero electroporated cells.

Most cortical folding studies employ the gyrification index (GI), which the authors have also used previously, but GI is not included here, for reasons that are unclear. This should be straightforward to include, since contralateral non-electroporated cortex is present in every section. If possible, it would be most beneficial to the field to include GI among the metrics used.

sFGFR3 IUE generates a thinner cortex with fewer oRG, IPs, and upper layer neurons, which alone creates an interpretive issue with regard to normal gyrification/sulcation – cytoarchitechtonically-defined cortical areas (e.g. individual Brodmann areas) typically span multiple sulci. Current findings seem entirely consistent with a selective effect on gyral "bulging" rather than sulcal formation per se, but this would depend on whether sufficient EP occured in sulcal regions. The electroporations themselves span sulci, but most of the construct-bearing (green) cortical mantle cells – and perhaps more so in controls? – appear to be in gyral crowns rather than sulcal depths. Most folding mechanisms require a local difference or asymmetry between bulging gyri and indenting sulci, but sulcal-gyral differences in electroporation and phenotype are not delineated here. In addition to differences across this tangential domain, differences in radial distribution of green cells could be important, given the non-autonomy of sFGFR3, but this is also not assessed. In general, careful interpretation of current findings in relation to existing models of cortical folding would be essential to fully realize the impact of this work.

The authors main point, that expression varies by 'region' is understood, but it needs to be shown that these changes across the tangential axis of developing cortex are meaningful for the topic at hand. Therefore, this data would become more interesting if the authors were to show another relevant molecule(s) that are expressed by neural precursor cells that do *not* show lateral to medial gradients of expression.

The authors show that most *FGFR1*, 2 and 3 cells are Pax6 positive and Tbr2 negative, and conclude that these cells must therefore be translocating radial glia, but it would be good to recognize that astroglial cells express Pax6 during and after neurogenic stages of development, therefore some of the cells could be related to gliogenesis not neurogenesis (an important point that is often lost in studies of cortical development, we all sometimes lose sight of the fact that cortex contains many many glial cells too).

---

## [Author Response]

Essential revisions:Given that the authors recognize non-cell autonomous role of FGFR3 in regulating FGF signaling,the authors should perform in situ hybridization for a few of the downstream effectors of FGF signaling and correlate the phenotype with in utero electroporated cells.

In accordance with the reviewer's comment, we performed *in situ* hybridization for a downstream effector of FGF signaling. We found that Sprouty2 expression, which is known to be increased by the activation of the FGF pathway, was markedly suppressed by sFGFR3 even in GFP-negative non-transfected cells. This result clearly indicates that sFGFR3 works non-cell-autonomously. This result was newly added as Figure 1—figure supplement 1 and was written in the text (Results section, paragraph one).

Most cortical folding studies employ the gyrification index (GI), which the authors have also used previously, but GI is not included here, for reasons that are unclear. This should be straightforward to include, since contralateral non-electroporated cortex is present in every section. If possible, it would be most beneficial to the field to include GI among the metrics used.

As suggested by the reviewer, we measured the local GI values. As in the case of other quantifications, to minimize the variation of the local GI values depending on the positions of coronal sections in the brain, the local GI value on the electroporated side and that on the contralateral non-electroporated side of the cerebral cortex in the same brain sections were measured, and the former was divided by the latter (local GI ratio). Consistent with other values, the local GI ratio was significantly decreased by sFGFR3. This result was added as the new Figure 1 and was written in the text (Results section, paragraph two). We also added a new Figure 1—figure supplement 4, which included the definition of the local GI ratio.

sFGFR3 IUE generates a thinner cortex with fewer oRG, IPs, and upper layer neurons, which alone creates an interpretive issue with regard to normal gyrification/sulcation – cytoarchitechtonically-defined cortical areas (e.g. individual Brodmann areas) typically span multiple sulci. Current findings seem entirely consistent with a selective effect on gyral "bulging" rather than sulcal formation per se, but this would depend on whether sufficient EP occured in sulcal regions. The electroporations themselves span sulci, but most of the construct-bearing (green) cortical mantle cells – and perhaps more so in controls? – appear to be in gyral crowns rather than sulcal depths. Most folding mechanisms require a local difference or asymmetry between bulging gyri and indenting sulci, but sulcal-gyral differences in electroporation and phenotype are not delineated here. In addition to differences across this tangential domain, differences in radial distribution of green cells could be important, given the non-autonomy of sFGFR3, but this is also not assessed. In general, careful interpretation of current findings in relation to existing models of cortical folding would be essential to fully realize the impact of this work.

As exemplified with Figure 1, GFP-positive areas included both the gyrus and the sulcus. Furthermore, we confirmed that GFP-positive cells were evenly distributed in the transfected areas at E40, when the gyrus was not yet formed (Author response image 1). Therefore, it seems unlikely that electroporation predominantly introduced sFGFR3 into the gyral regions. We added this point to the text (Results section, paragraph two).

As suggested, we described the laminar distribution of GFP-positive cells, which were almost the same between control and sFGFR3 samples, in the text (subsection “Upper layers are predominantly affected by FGF signaling”) and incorporated this result as the new Figure 5—figure supplement 1.

We agree with the reviewer that it would be important to include the discussion about the relationship between our findings and existing models, and we added it to the Discussion section.

**Author response image 1. respfig1:** Distribution of GFP-positive cells in the sFGFR3-transfected ferret brain at E40. pCAG-EGFP and pCAG-sFGFR3 were electroporated at E33. The coronal sections were prepared at E40 and stained with Hoechst 33342 (blue). GFP-positive cells were widely distributed in the ferret cerebral cortex. Scale bar = 2 mm.

The authors main point, that expression varies by 'region' is understood, but it needs to be shown that these changes across the tangential axis of developing cortex are meaningful for the topic at hand. Therefore, this data would become more interesting if the authors were to show another relevant molecule(s) that are expressed by neural precursor cells that do *not* show lateral to medial gradients of expression.

As suggested by the reviewer, we performed immunostaining for Pax6, which is a neural progenitor marker, and compared the expression patterns of neural progenitors with those of FGFRs. We found that these expression patterns were similar, suggesting that the regional difference of FGFR expression reflects that of neural progenitors. Our results suggest that FGF signaling induces a preferential increase of FGFR-positive neural progenitors in the prospective gyrus regions. In addition, our finding that the regional differences of FGFR expression found in the OSVZ in ferrets were not observed in mice (see Minor point #4) also supports our idea that the regional differences of FGFR expression are important for cortical folding. These results were incorporated into Figure 2—figure supplement 1 and were written in the Results section.

The authors show that most FGFR1, 2 and 3 cells are Pax6 positive and Tbr2 negative, and conclude that these cells must therefore be translocating radial glia, but it would be good to recognize that astroglial cells express Pax6 during and after neurogenic stages of development, therefore some of the cells could be related to gliogenesis not neurogenesis (an important point that is often lost in studies of cortical development, we all sometimes lose sight of the fact that cortex contains many many glial cells too).

As pointed out by the reviewer, it was previously reported that GFAP-positive astrocytes express Pax6 in mice (Journal of Neuroscience, 2008, 28, 4604-4612). Therefore, although our result showed that FGFRs were preferentially expressed in Pax6-positive/Tbr2-negaitve cells of the OSVZ in the developing ferret cerebral cortex, it remained possible that these cells were astroglial cells. To test this possibility, we examined if Pax6-positive cells in the OSVZ expressed GFAP. We found that Pax6-positive cells in the OSVZ did not express GFAP, suggesting that they are not astroglial cells. These results were added as the new Figure 2—figure supplement 4 and were written in the Results section.